# The Development and Validation of a Scale to Measure University Teachers' Attitude towards Ethical Use of Information Technology for a Sustainable Education

**Liliana Mâță** [1] , **Otilia Clipa** [2,*] and **Katerina Tzafilkou** [3,*]

1    Teacher Training Department, Vasile Alecsandri University of Bacău, 600115 Bacau, Romania;
     liliana.mata@ub.ro
2    Educational Sciences Department, Ștefan cel Mare University of Suceava, 720029 Suceava, Romania
3    Department of Information System, University of Macedonia, 54006 Thessaloniki, Greece
*    Correspondence: otiliac@usv.ro (O.C.); tzafilkou@uom.edu.gr (K.T.);
     Tel.: +40-7448128886 (O.C.); +30-2310891768 (K.T.)

**Abstract:** A self-administered measurement tool was developed and validated to provide data regarding ethical attitude of university teachers for a sustainable education. The research was based on several factors involved in forming attitudes towards the unethical information technology use. The sample: 334 Romanian teachers' respondents who teach in the higher education system contributed to this research. A successions of factor analyses and structural equation modeling showed that a second-order model is a good fit for experimental data ($\chi2/df = 1.75$, comparative fit index = 0.958, root mean square error of approximation = 0.045), however the partial least square (PLS-SEM) CFA approach revealed higher scores of factor loadings, implying the best fit to the model. This research suggested a structural model of ethical attitude of university teachers, composed by four factors measured by 13 indices. The results demonstrated that PLS-SEM CFA is appropriate for creating a valid structural model to measure university teachers' ICT ethical attitude. The current research predicted a theoretical contribution to the field of ethical attitude of university teachers within a sustainable education context.

**Keywords:** CBA factor analysis; ICT ethics; PLS-SEM factor analysis; scale validation; teachers' ethical attitude

## 1. Introduction

The university has very important roles in education for sustainability [1] and for development of future active teachers and citizens, including cognitions, behavioral and attitudinal domain [2]. The purposes of the university training are in continuous (re)sizing and change, according to the transformation and challenges from society, with development of new technology, and they also determine the perspective modifications [3–5]. In the academic area, education for sustainability proposed purposes is various and implies the cognitive (critical thinking, anticipatory thinking, metacognitive abilities), as well as the affective and ethical formation, as the university could be considered the one which completes the formation of the human personality and accomplishes the personality of the next generation who wants to find the different ways of living for living better [6–10].

In the documents drawn up by the institutions of the UNESCO report and European Commission, there are guidelines which make it possible to harmonize the systems of education and to establish some action directions regarding the aim of sustainable development. These issues can be obtained through

the ethical use of information and communication technologies (ICT) in education for preparing a positive perspective for use with respect for limited resources and for an ethical and ecological attitude of new technology. In the UNESCO documents, it is specified that one of the better ways to improve quality of education is sustainable development, which must be implemented in all levels and all social contexts [1]. All these reports underline important issues such as: the progress for realizing lifelong learning and education for all, the balance between fairness and excellence in education, training of people for ICT master and use, enhance the use of enabling technologies in particular ICT [11,12]. This document [1] underlines the necessity of developing cross-cutting competence (transversal competence) as: systems thinking competency, integrated problem-solving competency, and normative competency, which is the ability to transform the value in action. These competences are represented as an aggregate of information, cognitive skills and attitudes of learning contexts with ICT tools [13–15] and became an important part of delivering the modern university curriculum for training in all fields, but especially for teacher education [9,16]. The actual curriculum is combined between pedagogical approaches [17] and ethical competences regarding using ICT by university teachers and they are the powerful factor for transforming the next generation for sustainable development [2]. We note that between these complex competences, there is also a complex digital competence training involving the adaptation of a certain epoch, where most of the activities are carried out through assistance with technical means [6,18].

We could notice that the experts took into consideration the necessity of training skills to work with technology and they have the ability to integrate it into their training and life with the appropriate values and skills needed for effective human–machine collaboration in life [9,19], learning and teaching [5,17,20], work [11,21], and for sustainable development [12]. The developments of ICTs are a high impact for functionality of computer programs in an informational society and for sustainability of environments [22,23]. The key competences should be formed through both initial and continuous training for teachers [5,15] and the university teachers increase the academic achievement of students [6] and positive and responsible attitude for using electronically tools [17]. This involved adapting university education through curricular reform and continuous training of those who can be actors of the university education: teachers, students, and others academic persons [14,24]. Implementation of ICT in educational settings such as universities could bring out the pedagogical point of view with some question marks: "what is the added value of technology in learning?; What is ethical and unethical use of ICT? [4,11,25–27]. This is a challenge for social, cultural level and supposes an ethical attitude [10,28,29]. Regarding the moral attitude of using ICT, the dishonesty is considerable high in the academic field [22,26,27,30–32]. With the advance of computer technology, the terms of the moral field involved new concepts in addition to cheating and plagiarism [26,27,30,33]. The terminology is changing and it refers to ethics of computer use and are very important issues for legacy, for respect of power and functionality of computer programs in an informational society and for sustainability of environments [22,23]. The most important challenges for ethical attitude of technology use in university are [34]: digital identity in education, critical and judicious use of information, ICT-related abuse and online security and privacy, in school and family contexts, intellectual property in this context, dissemination of information and the sharing of knowledge on the Internet.

The university must discover and proactively identify this process to increase academic integrity through strict procedures and punishments and, at the same time, develop the culture of academic responsibility and good faith for using virtual tools [33]. In a document of research "Mapping major changes to education and training in 2025", Stoyanov et al. [35] declared that in the future (next 5 years), the major changes in the education field will be in rapport between formal and informal training and integration of learning styles with the new technologies and how to deliver these information [17]. These challenges for pre-service and in-service professional training determined very different attitudes from positive, such as enthusiasm, enjoyment, satisfaction, flow [36,37] to anxious—stress, frustration, fear, experience feeling of discomfort [6,14,38,39]. This attitude about innovation in the technology field influences the educational process at cognitive or moral levels and the use of ICT depends on this

attitude for using electronical tools [5,6,13,14,24,25,40]. Charki et al. [41] proposed, as a sustainable and mitigating solution, the legal intervention by influencing the cost-benefit analysis in determining the decision to commit unethical use of information technology.

The examination indicated that the existing studies only tackled ethical attitudes, identifying challenges in ICT usage and teaching practices by university teachers. None of the mentioned studies have employed the factor analysis methodology to develop and validate a common scale for ICT ethical attitude factors. Drawing from the above, this study develops and validates a scale to measure university teachers' ethical attitude towards ICT usage. A measurement and structural model are proposed according to the literature and the hybrid factor analysis methodology (is traced. This research is innovative, since it includes both a partial least square (PLS-SEM) and a covariance-based SEM (CB-SEM) approach using different software, to develop and validate the suggested model. Finally, group-based differences (gender, age, specialization, and frequency of IT use) are examined across the four extracted factors of university teachers' ICT ethical attitude.

## 2. Theoretical Background

### 2.1. Attitude Towards Unethical Information Technology Use

Attitude significantly affects a person's intention to behave ethically or unethically [42]. Defining the notion of attitude towards the unethical use of information technology is difficult, as there is confusion and lack of clarity in regards to the conceptualizing of ethics in the field of digital resources [43]. If, at a general level, the attitude refers to "the evaluation by the individual of how favorable an unfavorable act is" [44], in a specific aspect, the attitude towards the unethical information technology use aims at the appreciation by the individual of what is right or wrong in the case of digital instruments vehicles. The unethical information technology use (UITU) is defined by Chatterjee [45] as "the violation of privacy, property, accuracy, and access of any individual, group, or organization by any other individual, group, or organization". Privacy, accuracy, property, and access are the four ethical aspects of the information age discovered by Mason [46]. Other authors [47] identified five main factors of attitudes towards computer use: ownership, access, motivation, responsibility, and confidentiality.

The ethical attitude towards UITU is dynamic, because it depends on "the evaluated situation and changes as society changes" [48]. The formation of the attitude towards the use of information technology depends on many factors, both internal, which concern the personal values, the system of beliefs, as well as external, which refer to the social environment, the legal environment, etc. Therefore, it is very important to explore these attitudes over time to observe that the factors influencing their formation could change. As a result of the increasingly frequent use of information technology in the academic environment, it is all the more important to investigate the ethical attitudes of teachers.

### 2.2. Unethical Attitudinal Model in the Context of IT Use

The attitude depends on the individual characteristics of the people, on the moral development [49], on the beliefs of the individual, or on the judgments regarding the ethics of an act [50]. Two basic theories of attitudes have been harnessed to create the new model of unethical attitude towards the UITU: the technology acceptance model [51] and the cognitive-affective model [52]. At the base of the elaboration of the new approach is the model of acceptance of the technology, according to which the attitude is influenced by the beliefs of a person regarding the utility of the technology and the ease of use.

The cognitive component of the ethical attitude is an evaluation of concrete situations in higher education, which is the belief or disbelief of the teacher about the use of information technology. From a cognitive point of view, the attitude includes a storage section in which an individual organizes useful information on ethical aspects of the use of technological resources. The affective component is the emotional response manifested by pleasure or displeasure regarding the ethical aspects of

using information technology. The attitude towards unethical information technology use cannot be determined by simply identifying the beliefs, because the emotion works simultaneously with the cognitive process. As Agarwal and Malhotra [53] pointed out, emotions and faith-based evaluative judgment are mixed to obtain an integrated model of attitude. The behavioral component involves the favorable or unfavorable response regarding the action of ethical use of information technology. Jain [54] emphasized that the degree of consistency is different in the occurrence of attitude responses. At the behavioral level, the intention of ethical use of information technology comprises two sub-components: on the one hand, it aims at the acquisition of information technology, and on the other hand, it refers to the transmission and manipulation of data.

At the base of the theoretical model of the attitude towards the unethical use of information technology are the three components, cognitive, affective, and behavioral. The behavior of university teachers regarding the use of information technology is determined by the level of training and development of each component of attitudes, as well as the correlation and articulation of these components.

## 3. Materials and Methods

### 3.1. Research Model

The development of the model was based on experimental validated theoretical concepts developed in other research [55]. A four-dimensional model (Figure 1) was proposed for this research. The attitude was measured using a multidimensional construct (first and second order) with four specific dimensions of first order, which were manifestations of it. The multidimensional and hierarchical representation of university teachers ' attitude towards the UITU is primarily based on theoretical arguments, as outlined in the previous paragraphs.

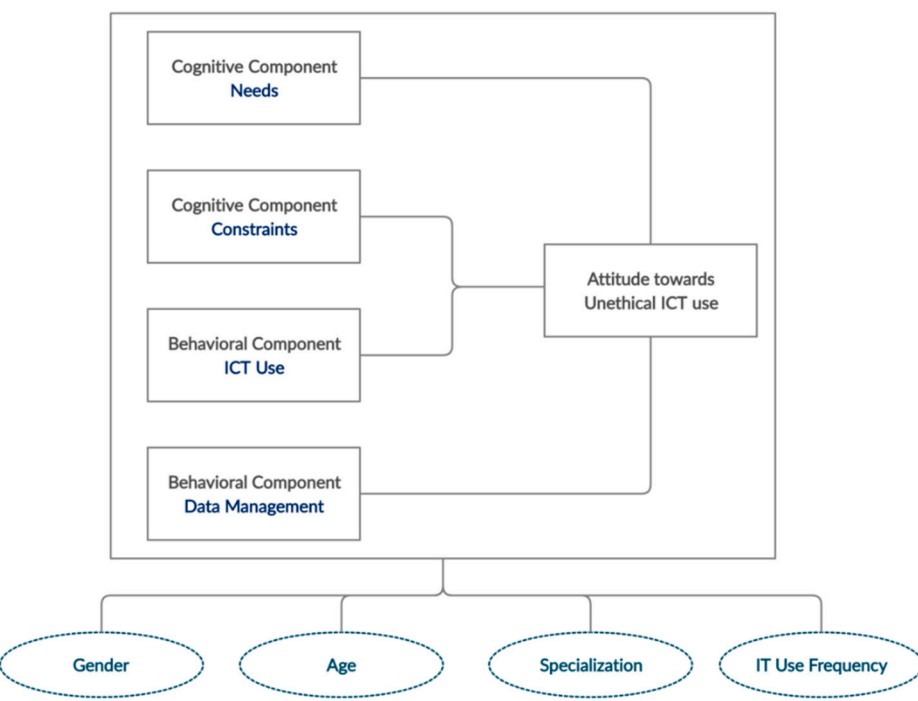

**Figure 1.** (Un)Ethical attitudinal model in the context of information and communication technologies use by university teachers.

The new model is analyzed in correlation with the factors that influence the attitude towards the unethical use of information technology. Regarding the unethical use of information technology in the academic environment, the attitude of university teachers is influenced by a number of individual

factors, such as gender, age, but also by a number of external factors, such as the specialization, scientific title, academic degree, the frequency of use of information technology during the courses and seminars.

### 3.1.1. Gender

Several authors [22,56–65] have observed that gender, as an individual factor, could be an important indicator of the ethical attitude regarding the use of information technology. The results of the studies showed that the female subjects adopted a more ethical attitude when using information technology than the male ones. According to the results of the research conducted by Akdemir et al. [66], the probability of the occurrence of ethical behaviors in virtual environments is higher in men than in women, as opposed to the real environment, in which no difference was found. There are studies [15,67,68] that have shown that gender does not have a significant influence on ethical behaviors.

### 3.1.2. Age

In some research [56,69,70], it has been shown that age is an important factor in ethical decision making and that older people are less influenced by external factors. The results of some studies [71] have shown that there is no interaction between age and ethical use of information technology.

### 3.1.3. Specialization

The ethical response at the cognitive, affective, and behavioral level may be different for university teachers depending on their specialization. There are studies [25,58,60,72,73] in the field of the use of information technology that have explored the influence of specialization on ethical attitude or behavior. The analysis of previous research has led to the finding that specialization was a factor investigated only in students in higher education. For specialization and ICT roles in teaching and learning in university, there are positive relations between study field and computer experience [5] and technical and humanistic domain [7,22,29,74].

### 3.1.4. Frequency Use of Information Technology

While the results of some studies [32,56,58,73,75,76] have shown that frequent use of information technology influences ethical behavior, data from other studies [37] have shown that the time of using digital resources does not influence a person's decision-making process.

### 3.2. Participants

Research was carried out on the basis of the application of a questionnaire developed in the context of thorough research on the subject of ethics. In the period July and October 2019, the questionnaire was administered online. The questionnaire items were measured on a Likert scale with 5 degrees of intensity (1 "total disagreement", 7 "total agreement"). The questionnaire asked for information on the profile of teachers (gender, age, scientific title (e.g., PhD student, assistant, lecturer, associate professor, professor), highest academic degree (e.g., master, PhD, postdoctoral), IT usage frequency during course (e.g., never, only a few times per semester, once per module/chapter, on each course, IT usage frequency during seminar (e.g., never, only a few times per semester, once per module/chapter, on each course), specialization (e.g., Arts, Communication Sciences, Economic Sciences and Business Management, Educational Sciences, Engineering, History and Cultural Studies, Informatics, Information Technology, Kine to therapy, Legal Sciences, Philology, Philosophy, Physical Education and Sport, Political and Administrative Sciences, Psychology, Science, Sociology, Geography). In this study, only part of the influence of the variable categories on the attitude of teachers on the ethics of IT use in higher education, namely the gender category, was examined. Of the 1500 emails sent, 380 replies were received, which were analyzed for understanding the perception of the unethical use of IT. After eliminating the answers appreciate being outliers resulted in a working sample (N = 334). The respondents' average

age was 45 years old (min = 20, max = 70, stdev = 9.6) and most of them were women. Respondents' socio-demographic characteristics are presented in Table 1.

**Table 1.** Respondents' socio-demographic characteristics (N = 334).

| Gender | n% | Academic Title | n% | Specialization | n% | Frequency of IT Use | n% |
|---|---|---|---|---|---|---|---|
| Female | 62.6% | Assistant | 12.0% | Informatics | 11.4 | Very often | 65.9 |
| Male | 37.4% | Lecturer | 36.8% | Education Sciences/ Psychology | 21.0 | Quite often | 11.7 |
| | | Associate Professor | 30.2% | Philology | 15.9 | Pretty rare | 17.1 |
| | | Professor | 18.66% | Physical Education/ Physical Therapy | 3.9 | Never | 5.4 |
| | | PhD Student | 2.4% | Engineering | 12.9 | | |
| | | | | Sciences | 13.5 | | |
| | | | | Sociology/ Political sciences | 10.5 | | |
| | | | | Economics | 11.1 | | |

## 3.3. The Survey Instrument

The initial scale comprised 35 items distributed on 4 dimensions, consistent with the ethical attitude model towards the use of information technology (IT).

- Dimension 1: Associated with the cognitive component-needs.
- Dimension 2: Associated with cognitive component-constraints.
- Dimension 3: Associated with behavioral component-acquisition of computer technologies.
- Dimension 4: Associated with behavioral component-data management.

For the needs factor associated with the cognitive component, in Dimension 1, 17 items were developed, of which items CN1, CN2, CN7 were adapted after Sondhi [77], the item CN5 after Gregory and Noto [78], the CN6 item after Pérez-Rodríguez et al. [79], items CN9, CN10, CN11, CN12, CN13, CN14, CN15, CN16, CN17 after Hashim and Hassan [80], and items CN3, CN4, CN8 are original. For the constraints factor associated with the cognitive component, within dimension 2, 6 items were developed, of which all items CC1, CC2, CC3, CC4, CC5, and CC6 are original. Regarding dimensions 3 and 4, the behavioral component (acquisition of information technologies and data management), we formulated 15 items, of which the BA1, BA2, BA8, BA12 are adapted after Namlu and Odabasi [81], items BA3, BA5, BA7, BA11, BA13, BA15 are original, the BA4 item is adapted after Hashim and Hassan [80], the item BA6 after Ozair [82] and the items BA9, BA10, BA14 after Etter et al. [83]. Items are renamed to BIT and BDM after the confirmation of the two factors they belong to, which are extracted from the exploratory factor analysis.

The questionnaire was used to assess teachers' attitudes towards the unethical use of information technology in higher education. Information technology involved physical resources (computers, laptops, and tablets), software resources (applications, educational software), virtual resources (web pages, e-mail, etc.), and telecommunication services (telephone or Internet) used to store, retrieve, transmit, and manipulate data in educational context. The completion of the questionnaire confirmed the agreement to participate in the research. This study respects the Helsinki Declaration on the Rights of Human Rights of Research Participants. The data were used exclusively for research purposes.

## 3.4. Analytical Procedures

In this research, the conditions for the applicability of multivariate analysis methods have been verified in accordance with the recommendations in the literature [84]. The general validity testing of the model was carried out in the framework of the SEM (structural equation modeling) approach by applying the CFA method of comparing alternative models: A CFA model of order one (measurement model) and a second-order CFA model (model structural). Our scale validation included also a

PLS-SEM analysis mainly because of the non normal distribution of the sample data, as suggested by Afthanorhan [85]. That is, in the order one model, we applied two iterations of CFA, one PLS-SEM approach, and one CB-SEM approach using different software: SmartPLS for the first and AMOS for the latter.

The testing of the one order model one included the following tests:

- analysis of the unidimensionality and internal consistency of the measurement scale;
- the testing of convergent validity;
- the testing of discriminatory validity.

The second order model was evaluated for the purpose of identifying and characterizing the links (associations) between the ordinal factor two and the other four factors of order one. The second-order test was conducted with the AMOS 26.0 version using the covariance matrix as input and the maximum verosimility method for estimating parameters.

In this study, the following indices for the quality assessment of a model are used: Tucker–Lewis index (TLI), comparative fit index (CFI), root mean square error of an (RMSEA), and root mean square residual (SRMR). For a model with acceptable quality, it was recommended (Ref) the following threshold values: RMSEA $\leq$ 0.08, SRMR $\leq$ 0. 08, TLI $\geq$ 0.95, CFI $\geq$ 0.95 [84]. For the purpose of testing the equivalence (invariance) of the model in the two groups of teachers (men and women), the method of multi-group confirmatory (MGCFA) factorial analysis was used in a succession of test levels. To examine for group-based potential differences cross the extracted factors, we used non parametric methods, because of the not normally distributed data of the sample [86].

## 4. Results

### 4.1. Exploratory Factor Analysis

Before conducting the exploratory factor analysis (EFA), we performed the Kaiser–Meyer–Olkin (KMO) test [87] to measure the sampling adequacy and the Bartlett's test of sphericity [88] to investigate the factorability of the data. As depicted in Table 2, KMO results of high value (almost 0.85) implied the suitability of the data for EFA and a significant test statistic was indicated by Bartlett's test of sphericity ($p < 0.001$).

**Table 2.** Kaiser–Meyer–Olkin (KMO) sample adequacy test.

| KMO and Bartlett's Test | | |
|---|---|---|
| **Kaiser-Meyer-Olkin Measure of Sampling Adequacy** | | 0.844 |
| Bartlett's Test of Sphericity | Approx. Chi-Square | 3,479,346 |
| | df | 703 |
| | Sig. | 0.000 |

In the EFA, all 35 items were subjected to principal component analysis (PCA) with Promax with Kaiser rotation. The initial analyses showed 11 factors with eigenvalue over 1, however, as depicted in the scree plot in Figure 2, there is a break after the third component, and several items showed small variances and close to each other.

A second iteration of EFA use was then performed and several items were removed, because they performed communality scores below 0.4 [89,90]. Then, a third extraction cycle was performed, clearly revealing four factors and the items performing lower than the 0.5 factor load [91] were also removed.

The final four-factor model with the remaining 14 items accounted for 62% of the total variance. The first factor called 'Cognitive Needs', with an eigenvalue of 1.40, included three items. The second factor called 'Cognitive Constraints, with an eigenvalue of 1.30, included three items. The third factor called 'Behavioral ICT', with an eigenvalue of 3.99, included three items, and the fourth factor called 'Behavioral Data Management', with an eigenvalue of 1.98, included four items.

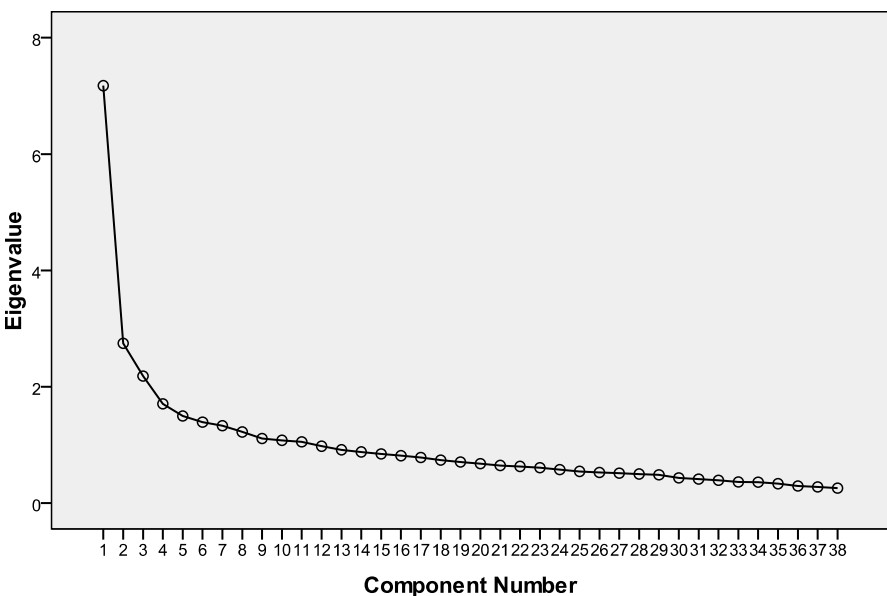

**Figure 2.** Scree plot output indicating that the data have four factors [89].

Table 3 below, presents the item wordings, factor loadings, eigenvalue, variance explained, and Cronbach's alpha for each factor. As depicted, the factor loadings ranged from 0.50 to 0.85, meaning that all items were good measures of their respective factors [91]. As indicated by Cronbach's alphas that were above the threshold value of 0.70, all factors were internally consistent and well defined by their items [92].

**Table 3.** Results of exploratory factor analysis (EFA) on the 13-item ICT ethical attitude scale.

| Factor/Item [1] | Factor Loading | Eigen Value | Variance Explained | Cronbach's Alpha |
|---|---|---|---|---|
| CN: Cognitive Needs | | 3.995 | 28.535 | 0.747 |
| CN1 | 0.805 | | | |
| CN2 | 0.832 | | | |
| CN3 | 0.789 | | | |
| CC: Cognitive Constraints | | 1.982 | 14.161 | 0.728 |
| CC1 | 0.846 | | | |
| CC2 | 0.812 | | | |
| CC3 | 0.712 | | | |
| BDM: Behavioral Data Management | | 1.411 | 10.007 | 0.709 |
| BDM1 | 0.776 | | | |
| BDM2 | 0.759 | | | |
| BDM3 | 0.720 | | | |
| BDM4 | 0.706 | | | |
| BIT: Behavioral ICT | | 1.298 | 9.274 | 0.783 |
| BIT1 | 0.795 | | | |
| BIT2 | 0.887 | | | |
| BIT3 | 0.660 | | | |

[1] All the items are measured on a 5-point Likert scale (1: strongly disagree to 5: strongly agree).

### 4.2. Confirmatory Factor Analysis

A PLS-SEM confirmatory factor analysis (CFA) was performed using the SmartPLS software, to establish the structural validity of the scale. By examining the absolute values of skewness and kurtosis as well as the normal distribution of the data, we observed an approximately non normal distribution [86] cross all the measured variables. Hence, PLS-SEM was chosen for the primary iteration of the confirmatory analysis mainly because it is appropriate for non normally distributed data [93], and is suitable for complex models with numerous endogenous and exogenous constructs and indicator variables [93,94]. In addition, our sample size exceeds the recommended value of 50, i.e., 10 times larger than the number of items for the most complex construct [95].

The EFA extracted four-factor PLS-SEM model comprises of unidirectional predictive relationships between each of the latent construct that is linked with the observed indicator [91]. PLS-SEM results suggested a good fit of the model (SRMR = 0.075, NFI = 0.726) according to the criteria of acceptance suggested in Bryne [96], Hair et al. [91], and Kline [97]. In addition, by examining the latent variable correlations, the results showed that significant correlations existed between the factors ($p < 0.01$).

As depicted in Figure 3, the loading factors' performance comply with Awang [98] recommendation of at least 0.5. In addition, t values (depicted in the constructs' relationship paths) and p values are all accepted and significant. Additionally, Table 4 depicts that all composite reliability (CR) values are above 0.7, indicating internal consistency [99]. All average variance extracted (AVE) are above 0.5, indicating convergent reliability [100]. Finally, the values Rho_A reliability coefficients are all above 0.7, complying with the suggestions of Dijkstra and Henseler [101].

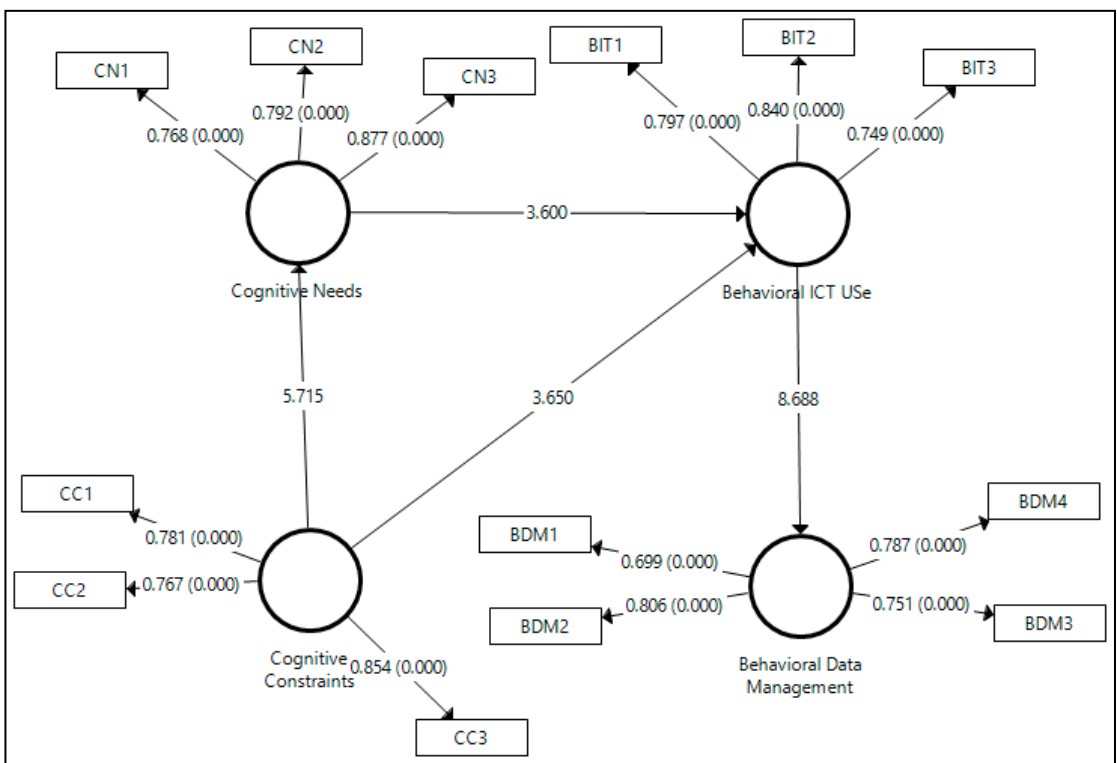

**Figure 3.** PLS-SEM confirmatory factor analysis and structural model of the teachers' ICT ethical attitude, with SmartPLS.

**Table 4.** Reliability, validity, and internal consistency results for the CFA SEM measurement model.

| Factor/ Item | CR | AVE | Rho_A |
|---|---|---|---|
| CN: Cognitive Needs | 0.853 | 0.660 | 0.765 |
| CN1 | | | |
| CN2 | | | |
| CN3 | | | |
| CC: Cognitive Constraints | 0.844 | 0.631 | 0.757 |
| CC1 | | | |
| CC2 | | | |
| CC3 | | | |
| BDM: Behavioral Data Management | 0.837 | 0.644 | 0.708 |
| BDM1 | | | |
| BDM2 | | | |
| BDM3 | | | |
| BDM4 | | | |
| BIT: Behavioral ICT | 0.845 | 0.525 | 0.794 |
| BIT1 | | | |
| BIT2 | | | |
| BIT3 | | | |

The discriminant validity was assessed using Fornel and Larcker [102] by comparing the square root of each AVE in the diagonal with the correlation coefficients (off-diagonal) for each construct in the relevant rows and columns. As depicted in Table 5, this measurement model supports the discriminant validity between the constructs.

**Table 5.** Discriminant validity.

| | Cognitive Needs | Cognitive Constraints | Behavioral Data Management | Behavioral ICT |
|---|---|---|---|---|
| Cognitive Needs | 0.813 | | | |
| Cognitive Constraints | 0.356 | 0.802 | | |
| Behavioral Data Management | −0.265 | −0.287 | 0.795 | |
| Behavioral ICT | −0.240 | −0.203 | 0.368 | 0.725 |

A replication of CFA was conducted based on the covariance-based SEM (CB-SEM) approach and using the AMOS 26.0 software and the maximum likelihood estimation. The CB-SEM analysis validated the fitness of the model: $\chi 2/df$ = 1.64, probability level = 0.001, comparative fit index (CFI) = 0.965, the Tucker–Lewis fit index (TLI) = 0.946, and RMSEA = 0.042 [103–107].

However, as depicted in Figure 4, the CB-SEM CFA revealed valid scores of factor loadings (>0.05) but several meet lower scores compared to the loadings generated through the PLS-SEM analysis. In that case, we perceive the PLS-SEM extracted model as the more accurate one according to the findings of Afthanorhan [85] who conducted a cooperative CBA analysis using both SmartPLS and AMOS software and concluded that PLS-SEM path modeling using SMARTPLS is appropriate to carry on the confirmatory factor analysis which is more reliable and valid. As the author explains, the lower factor loading scores in the CBA output, show that the PLS-SEM method is more appropriate to maximize the explained variance of endogenous latent constructs (dependent variable) and minimize the unexplained variances.

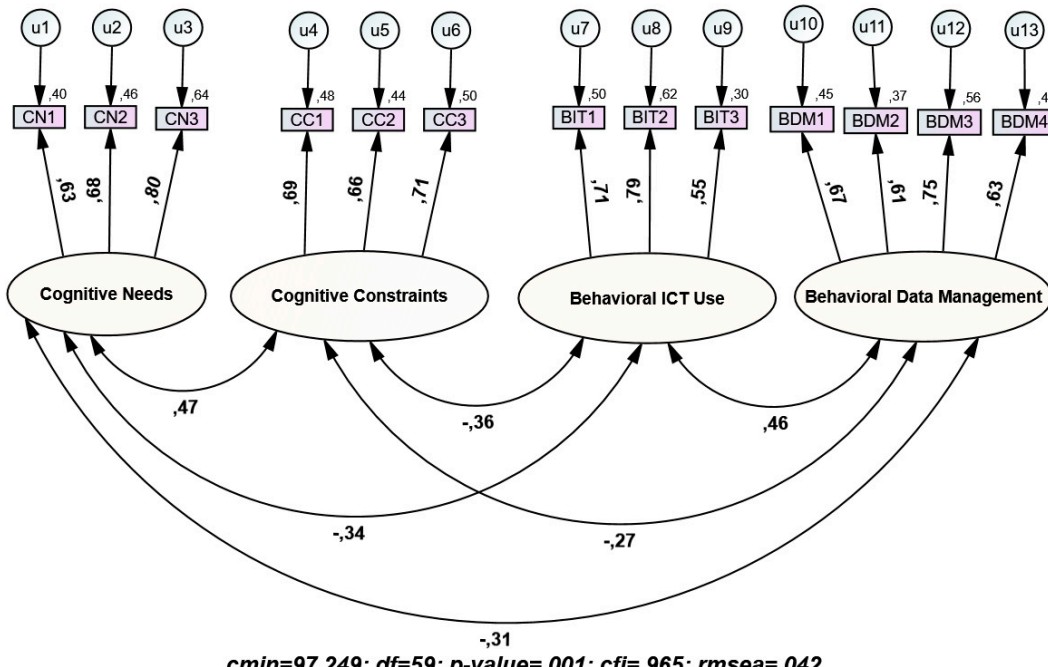

cmin=97,249; df=59; p-value=,001; cfi=,965; rmsea=,042

**Figure 4.** CB-SEM confirmatory factor analysis of the teachers' ICT ethical attitude with AMOS.

Unstandardized and standardized parameter estimates are all provided in Table 6 and are statistically significant at the alpha level of 0.001. The R2 values show the amount of variance of the items explained by the respective constructs.

**Table 6.** Results of CFA of the 13-item ICT ethical attitude scale (Appendix A).

| Item | Unstandarized Estimate | Standarized Estimate | $R^2$ | S.E. | C.R. | *p* |
|------|------------------------|----------------------|-------|------|------|-----|
| CA14 | 1.000 | 0.635 | 0.40 | | | |
| CA15 | 1.560 | 0.679 | 0.46 | 0.170 | 9.193 | *** |
| CA17 | 1.529 | 0.799 | 0.64 | 0.163 | 9.410 | *** |
| SA2 | 1.000 | 0.690 | 0.48 | | | |
| SA3 | 1.000 | 0.663 | 0.44 | 0.113 | 8.867 | *** |
| SA4 | 1.075 | 0.706 | 0.50 | 0.119 | 9.032 | *** |
| BA1 | 1.000 | 0.709 | 0.50 | | | |
| BA2 | 1.232 | 0.790 | 0.62 | 0.131 | 9.437 | *** |
| BA4 | 0.849 | 0.547 | 0.30 | 0.104 | 8.179 | *** |
| BA9 | 1.000 | 0.674 | 0.45 | | | |
| BA10 | 0.641 | 0.611 | 0.37 | 0.072 | 8.930 | *** |
| BA11 | 0.957 | 0.749 | 0.36 | 0.095 | 10.084 | *** |
| BA12 | 0.773 | 0.629 | 0.40 | 0.085 | 9.126 | *** |

*** = the values are statistically significance.

### 4.3. Second Order Confirmatory Analysis

A second order confirmatory factor analysis (CFA) was evaluated with the AMOS 26.0 version using the covariance matrix as input and the maximum verosimility method for estimating parameters. The second order analysis was conducted to test whether the four factors belonged to a single broader latent factor of teachers' ICT ethical attitude. The second order CFA results suggested a good fit according to the criteria suggested in Muthen and Muthen [107] and Bandalos [103], as depicted in Figure 5. In particular, χ2/df = 1.75, the p value is significant, the CFI is above 0.95, and the RMSEA is below the threshold of 0.80.

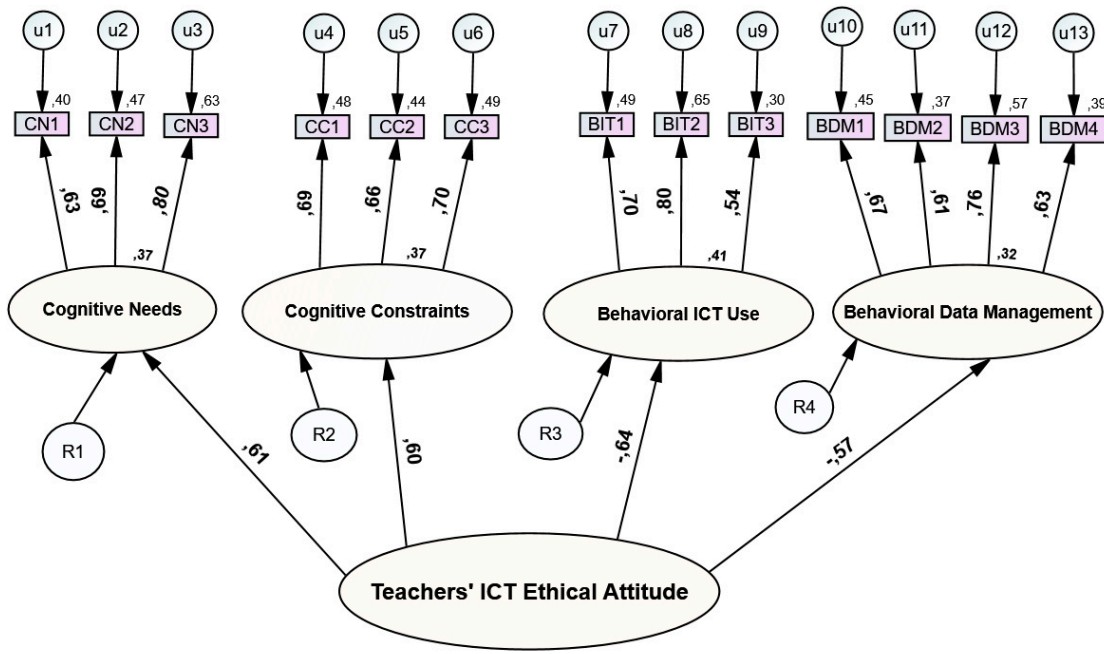

*cmin=107,113; df=61; p-value=,000; cfi=,958; rmsea=,045*

**Figure 5.** Model diagram of the 2nd order confirmatory factors of the teachers' ICT ethical attitude.

*4.4. Group Differences*

This study also examined the potential gender, age, specialization, and frequency of IT use differences in the means scores cross the four factors. As depicted in Tables 7–10, gender showed some significant differences in the factor of data management, while age and frequency of IT use showed a correlation with the factors of cognitive constraints and behavioral ICT use. Specialization revealed no significant differences for the examined sample.

**Table 7.** Mann–Whitney U test, grouping variable: gender.

|  | **CN** | **CC** | **BIT** | **BDM** |
|---|---|---|---|---|
| Mann–Whitney U | 12,546,500 | 12,596,000 | 12,261,500 | 11,576,000 |
| Wilcoxon W | 20,421,500 | 20,471,000 | 20,136,500 | 33,521,000 |
| Z | −0.660 | −0.407 | −0.943 | −1.984 |
| Asymp. Sig. (2-tailed) | 0.509 | 0.684 | 0.345 | 0.047 |

**Table 8.** Kruskal Wallis test, grouping variable: the main field of specialization.

|  | **CN** | **CC** | **BIT** | **BDM** |
|---|---|---|---|---|
| Chi-Square | 10.935 | 8.905 | 6.751 | 8.786 |
| df | 7 | 7 | 7 | 7 |
| Asymp. Sig. | 0.141 | 0.260 | 0.455 | 0.268 |

**Table 9.** Spearman correlation test between age and the four factors of university teachers' ICT ethical attitude.

|  |  |  | **Age** | **CN** | **CC** | **BIT** | **BDM** |
|---|---|---|---|---|---|---|---|
| Spearman's rho | Age | Correlation Coefficient | 1.000 | 0.064 | 0.144 [1] | −0.138 [2] | −0.059 |
|  |  | Sig. (2-tailed) |  | 0.243 | 0.009 | 0.012 | 0.282 |
|  |  | N | 334 | 334 | 332 | 334 | 334 |

[1] Correlation is significant at the 0.01 level (2-tailed); [2] Correlation is significant at the 0.05 level (2-tailed).

**Table 10.** Kruskal Wallis test, grouping variable: frequency of IT use.

|  | CN | CC | BIT | BDM |
|---|---|---|---|---|
| Chi-Square | 8.850 | 6.287 | 3.386 | 9.077 |
| df | 3 | 3 | 3 | 3 |
| Asymp. Sig. | 0.031 | 0.098 | 0.336 | 0.028 |

## 5. Discussion

The main objective of this study was to develop and validate a scale to measure unethical attitude towards ICT use from university teachers. The four-factor and 13 items scale that was extracted through EFA, has been validated using a hybrid approach mainly because of the non normal distribution found in the data. Results interestingly indicate that a PSL-SEM CFA produced the best fit to the model, in terms of factor loadings, while the CB-SEM approach generates a good model fit, however, scoring lower values of factor loadings.

The findings are in accordance with the work of Afthanorhan [85] who predicted for this issue since most the value of factor loading obtained in CB-SEM was lower than PLS-SEM even author use the same scales when apply the unidimensionality procedure. As explained, PLS-SEM is more appropriate for a CFA where not normal distributions are met and also, its application is aimed to maximize the explained variance of the endogenous latent constructs and minimize the unexplained variances. Contrarily, the CB-SEM approach is used to evaluate focuses on goodness of fit, which is focusing on minimization of the discrepancy between the observed covariance matrix and the estimated covariance matrix [91]. For this reason, its application suggests that the prior theory is strong. Similar to, in our, the value of factor loadings/outer loadings in PLS-SEM is better than CB-SEM CFA for the university teachers' attitude towards the ethical attitude of ICT.

As expected, the second order model, implemented via a CB-SEM approach and the AMOS software, revealed a good fit of the model, pertaining to the lower scores (compared to PLS-SEM CFA) factor loadings. The final model is valid, and our results strongly suggest the implementation of PLS-SEM CFA for the validation of the suggested model. From the group-based differences that were examined, it was interesting that gender expressed significant differences only in one factor (data management), while it did not show to affect any one of the four factors. This finding is in accordance with previous works [15,67,68] that stated no significant differences towards the unethical use of ICT between the two genders, however further investigation is needed. Another important finding was the differences detected among different groups of ICT use frequency in the two factors of cognitive constraints and behavioral data management. This is in accordance with previous studies [32,56,58,73,75,76] outlining that frequent use of information technology influences ethical behavior.

The developed and validated scale can serve as a tool to evaluate the university teachers' ethical attitude towards the use of digital technologies and support the development of the academic responsibility and good faith for using virtual tools. In addition, by revealing the important items and constructs of the scale, this study sheds light on the policies that should be made from the university management and authorities to preserve or endorse the ethical attitude towards ICT use inside the higher education institutions.

Adequate training of university teachers in accordance with the ethical requirements of digital education is becoming one of the essential conditions for sustainable development. The training of the future teachers or professionals should be done in the light of sustainable development [1,11,12], with the emphasis on the formation of ethical attitudes towards the use of ICT. In this way, the digital resources will be used efficiently in accordance with human and environmental needs, both now and in the future. The manifestation of an appropriate ethical attitude towards the use of information technology indicates responsibility on the part of teachers and students in the academic environment.

## 6. Conclusions

This study brings theoretical and practical contributions by developing and validating a scale to measure unethical attitude towards ICT use from university teachers. The ethical attitudinal model in the context of information technology use by university teachers provides the conceptual basis for understanding the cognitive, affective, and behavioral components involved in the moral positioning towards the integration of new technologies. The four-factor generated model revealed a good fit, demonstrating that cognitive needs and constraints as well as behavioral ICT use and data management are adequate components to measure the university teachers' (un)ethical attitude towards the use of technology. The resulted 13 items showed valid factors loadings and high values of consistency and reliability through both CFA procedures (PLS-SEM and CB-SEM), reinforcing the validity of the model.

A number of limitations of the study can be highlighted. The generalization of results is not possible due to the specific context and the small number of university teachers who participated in the study. Nevertheless, research efforts should be expanded on testing and the scale in a variety of educational contexts or countries in order to enhance its robustness and flexibility. Another limitation of the research derives from the small number of items obtained from the statistical processing of the data, rendering possible difficulties to future researchers if item elimination is needed in their adjusted studies.

A future research direction will be to expand the research group to investigate the attitude towards the ethical use of information technology and among students, master students, and doctoral students. The application of the validated instrument on a sample of teachers from the pre-university education system would contribute to the comparative analysis of teachers' attitudes. In addition, further factors can be examined, like for instance, personality traits or social influence regarding the use of ICT. Along with these future research directions, two important components will be developed at the level of educational practice in the academic space. On the one hand, the emphasis will be on the training component of university teachers in the context of the ethical use of information technology. On the other hand, the introduction of the ethical component of the use of information technology at the level of the curriculum in higher education will be pursued.

**Author Contributions:** Conceptualization, L.M. and O.C.; methodology, K.T.; software, K.T.; validation, K.T.; formal analysis, K.T.; investigation, L.M. and O.C.; resources, L.M.; data curation, K.T.; writing—original draft preparation, L.M., O.C. and K.T.; writing—review and editing, L.M.; visualization, O.C.; supervision, L.M.; project administration, L.M.; funding acquisition, L.M. All authors have read and agreed to the published version of the manuscript.

**Funding:** This research was funded by a grant of Ministry of Research and Innovation, CNCS - UEFISCDI, project number PN-III-P1-1.1-TE-2016-0773, within PNCDI III.

**Conflicts of Interest:** The authors declare no conflict of interest.

## Appendix A

**Table A1.** Scale of attitudes towards the unethical use of information technology.

| Type | Items | Totally Disagree | Partially Disagree | Neutral | Partially Agree | Totally Agree |
|---|---|---|---|---|---|---|
| | **Cognitive needs** | | | | | |
| CN1 | It is necessary to select retransmitted messages. | | | | | |
| CN2 | Attribution of authorship, without his permission, violates the ethical rules of using IT. | | | | | |
| CN3 | Distributing information in online environment without indicating the source violates ethical rules for the use of IT. | | | | | |
| | **Cognitive Constraints** | | | | | |
| CC1 | The teacher must check whether or not the students use information technology ethically. | | | | | |
| CC2 | The guide on the ethical aspects of using IT is explained to students before starting activities. | | | | | |
| CC3 | Sanctions are required for non-compliance with the ethical use of IT by students. | | | | | |

**Table A1.** *Cont.*

| Type | Items | Totally Disagree | Partially Disagree | Neutral | Partially Agree | Totally Agree |
|---|---|---|---|---|---|---|
| **Behavioral ICT use** | | | | | | |
| BIT1 | "Crack" programs can be used to purchase and process data. | | | | | |
| BIT2 | An unlicensed educational software can be used in current educational activities. | | | | | |
| BIT3 | Images and content in the online environment may be used in current educational activities without the permission of the authors. | | | | | |
| **Behavioral Data Management** | | | | | | |
| BDM1 | Phrases can be processed from an online source, in research work, without mentioning the source. | | | | | |
| BDM2 | Papers can be purchased online and presented as original, if the form in which they are presented changes. | | | | | |
| BDM3 | It is not necessary to specify all the sources from where the information was taken, in the elaboration of a course support. | | | | | |
| BDM4 | A digital tool (eg software) can be presented to others as original if minor changes are made to the interface. | | | | | |

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
