# Peer review of "The Development and Validation of a Scale to Measure University Teachers’ Attitude towards Ethical Use of Information Technology for a Sustainable Education"

_sustainability, doi:10.3390/su12156268_

Round 1

Reviewer 1 Report

Dear authors,

In my opinion the paper contains an interesting statistical analysis.

The objective is defined correctly. The conclusion solves and explains the objectives set. The study sample is relevant.

However, the summary discusses other possible structural models that have not been used, this could be included in 3.4 analytical procedures or in the introduction.

I also recommend to review the publisher's format specifications.

The paper has a clear structure. In my opinion, you have done a good job.

Sincerely,

Author Response

Response to Reviewer 1 Comments

Following we describe the revised parts explaining how we addressed the reviewer’s suggestions for minor revisions.

All revisions in the manuscript are highlighted using green and blue font background colors.

Reviewer 1

-The summary (discusses other possible structural models that have not been used, this could be included in 3.4 analytical procedures or in the introduction)

-Reply: As the reviewer suggested we proofed read the abstract and we removed all unnecessary or not used terms. In particular, there was a ‘wrong’ acronym (CBA) instead of CB, it has been removed (the deleted text is not visible in the revised version).

Thank you for reviewing our work,

The authors

Reviewer 2 Report

Comments to authors:

This is a priority topic in the educational field, which has been addressed through a real good statistical tool, but which could perhaps be improved if a more powerful theoretical load is added, and, above all, practical applications should be extracted from the results.

Specific Comments:

  1. From a Evolutive psychological point of view, one cannot speak exclusively of “personality” and its development. Authors should use a more up-to-date cognitive-behavioral evolutionary psychology conceptual frameworks.
  2. Perhaps, in line with the current research, it would be advisable to comment on ethical behaviors based on prosocial and antisocial behaviors (which are not specified until the end of the manuscript, and just partially).
  3. There is also no comment on the Introduction of the social pressure concept, regarding the use of ICTs.
  4. It is evident that it could not be foreseen in the design of the manuscript and the data collection, but it must be taken into account that the outbreak of the COVID-19 pandemic has totally changed the landscape of the use of ICTs in education by teachers, managers and students.
  5. The methodology used and the data analysis is excellent.
  6. However, they rely on a single study (marginal, 55) to define the data collection instrument. It would be advisable to support more information and especially the theoretical sources that justify its use in connection with the objectives of the study.
  7. Discussion: It is very theoretical and excessively long, perhaps due to the “generality” characteristic of the factors found.
  8. No practical application of the information obtained can be observed, nor how can it be refined to offer solutions to the problem raised in the introduction.

Author Response

Response to Reviewer 1 Comments

Following we describe the revised parts explaining how we addressed the reviewer’s suggestions for minor revisions.

All revisions in the manuscript are highlighted using green and blue font background colors.

Reviewer 2

- From a Evolutive psychological point of view, one cannot speak exclusively of “personality” and its development.

-Reply: The word personality was deleted from the first sentence.

- ethical behaviors based on prosocial and antisocial behaviors ???

– Reply: To confirm the reviewer’s concern we further highlight that we  include ‘ attitude’ in the manuscript.

- social pressure concept, regarding the use of ICTs ???

Reply: Since these factors were not a subject of the current study, we clearly explain in the Discussions that they can be included in a future work.

- they rely on a single study (marginal, 55) to define the data collection instrument ???

- Reply: The development of the research tool is based on several studies, which are mentioned in the text [77, 78, 79, 80, 81, 82, 83];

- Discussion: It is very theoretical and excessively long, perhaps due to the “generality” characteristic of the factors found.

-Reply: Future research directions have been moved to the conclusions section.

- No practical application of the information obtained can be observed, nor how can it be refined to offer solutions to the problem raised in the introduction 

-Reply, examples of practical applications have been added in the Discussion section.

Thank you for reviewing our work,

The authors

Reviewer 3 Report

This article refers to the "Measuring ethical attitude of university teachers for a sustainable education", in which a method of evaluation and validation of an assessment instrument is developed. I find this article sensational, as well as being a line of research that is not exploited in the scientific literature. All parts of it are well elaborated. The theoretical framework is adequate in length and number of authors. The research procedure is adequate and very justified with authors. The statistical procedures presented in the results are adequate. The discussion is appropriate.

Even so, there are a number of recommendations that I make to authors to include in their manuscript:

Title: I think you should include the words "scale" or "instrument" in the title. This would facilitate the search by the authors in databases, such as Web of Science or SCOPUS.

2.- Citations in the text, many of which are not presented as they should be. An example of this is line 38, putting [6,7,8,9,10] when it should put [6-10]. Check the rest of the text.

I recommend the inclusion in the theoretical framework of the following reference https://www.mdpi.com/2071-1050/11/20/5613. I consider that it is very related to the subject matter.

I recommend including a conclusion section, in which the general conclusion of the study, the limitations of the study and future lines of research are introduced.

Finally, I would like to congratulate the authors for their work.

Author Response

Reply to Reviewer 3

Following we describe the revised parts explaining how we addressed the reviewer’s suggestions for minor revisions.

All revisions in the manuscript are highlighted using green and blue font background colors.

Reviewer 3

- I think you should include the words "scale" or "instrument" in the title.

 - Reply: The title has been changed, according to the important reviewer’s suggestion:

Old title: Measuring ethical attitude of university teachers for a sustainable education

New title: The development and validation of a scale to measure university teachers' attitude towards ethical use of information technology for a sustainable education

- Citations in the text, many of which are not presented as they should be. An example of this is line 38, putting [6,7,8,9,10] when it should put [6-10].

-Reply: Thank you for considering this, all citations are verified.

- I recommend the inclusion in the theoretical framework of the following reference https://www.mdpi.com/2071-1050/11/20/5613.

-Reply The reference was added at 23 number. The old reference was deleted.

Rodríguez-García, A.-M.; López Belmonte, J.; Agreda Montoro, M.; Moreno-Guerrero, A.-J. Productive, Structural and Dynamic Study of the Concept of Sustainability in the Educational Field. Sustainability 2019, 11, 5613.

- I recommend including a conclusion section, in which the general conclusion of the study, the limitations of the study and future lines of research are introduced.

-Reply:  There is introduced the Conclusion section with the general conclusions of the study, the limitations and future lines of research, as suggested by the reviewer.

Thank you for reviewing our work,

The authors

Reviewer 4 Report

Dear authors, it is advisable to add new relevant references on the object of study of the last three years 2017/18 and 20

Finally, it is advisable to develop a greater analysis of the conclusions obtained.

Author Response

Reply to Reviewer 4

Following we describe the revised parts explaining how we addressed the reviewer’s suggestions for minor revisions.

All revisions in the manuscript are highlighted using green and blue font background colors.

- it is advisable to develop a greater analysis of the conclusions obtained.

- Reply: There is introduced a new Conclusion section with the general conclusions of the study, the limitations and future lines of research, as suggested by the reviewer. 

Thank you for reviewing our work,

The authors